# Proteome profile of patients with excellent and poor speech intelligibility after cochlear implantation: Can perilymph proteins predict performance?

Martin Durisin[1]ᴼ, Caroline Krüger[1]ᴼ, Andreas Pich[2], Athanasia Warnecke[1,3]*, Melanie Steffens[1], Carsten Zeilinger[4], Thomas Lenarz[1,3], Nils Prenzler[1]‡, Heike Schmitt[1,3]‡

1 Department of Otolaryngology, Hannover Medical School, Hannover, Germany, 2 Core Facility Proteomics, Hannover Medical School, Hannover, Germany, 3 Cluster of Excellence of the German Research Foundation (DFG; "Deutsche Forschungsgemeinschaft") "Hearing4all", Hannover Medical School, Hannover, Germany, 4 BMWZ (Zentrum für Biomolekulare Wirkstoffe), Gottfried-Wilhelm-Leibniz University, Hannover, Germany

ᴼ These authors contributed equally to this work.
‡ NP and HS also contributed equally to this work.
* warnecke.athanasia@mh-hannover.de

**Data Availability Statement:** The data set is available publicly through this link: https://github.

## Abstract

Modern proteomic analysis and reliable surgical access to gain liquid inner ear biopsies have enabled in depth molecular characterization of the cochlea microenvironment. In order to clarify whether the protein composition of the perilymph can provide new insights into individual hearing performance after cochlear implantation (CI), computational analysis in correlation to clinical performance after CI were performed based on the proteome profile derived from perilymph samples (liquid biopsies). Perilymph samples from cochlear implant recipients have been analyzed by mass spectrometry (MS). The proteins were identified using the shot-gun proteomics method and quantified and analyzed using Max Quant, Perseus and IPA software. A total of 75 perilymph samples from 68 (adults and children) patients were included in the analysis. Speech perception data one year after implantation were available for 45 patients and these were used for subsequent analysis. According to their hearing performance, patients with excellent (n = 22) and poor (n = 14) performance one year after CI were identified and used for further analysis. The protein composition and statistically significant differences in the two groups were detected by relative quantification of the perilymph proteins. With this procedure, a selection of 287 proteins were identified in at least eight samples in both groups. In the perilymph of the patients with excellent and poor performance, five and six significantly elevated proteins were identified respectively. These proteins seem to be involved in different immunological processes in excellent and poor performer. Further analysis on the role of specific proteins as predictors for poor or excellent performance among CI recipients are mandatory. Combinatory analysis of molecular inner ear profiles and clinical performance data using bioinformatics analysis may open up new possibilities for patient stratification. The impact of such prediction algorithms on diagnosis and treatment needs to be established in further studies.

com/vianna-research/perilymph-proteins-performance-publication.git.

**Funding:** This work was supported by the DFG Cluster of Excellence EXC 2177/1 "Hearing4all".

**Competing interests:** The authors have declared that no competing interests exist.

**Abbreviations:** CI, cochlear implantation; MS, mass spectrometry; i.e, id est—meaning "that is to say"; RSLC, nano-flow ultra-high pressure liquid chromatography system; LTQ, linear ion trap mass spectrometer; LFQ, label-free quantification; GOA, Gene Ontology Annotations; GO, Gene Ontology; HP, hearing performance; ERK, extracellular regulated kinase; BDNF, brain-derived neurotrophic factor; SBP1, selen binding protein 1; GPx1, glutathion peroxidase; uPAR, urokinase receptor; ERK, extracellular regulated kinase; MAC, coordinates membrane attack complex; MPO, Myeloperoxidase; HOCl, hypochlorous acid; NHE, sodium–hydrogen exchanger; CFTR, cystic fibrosis transmembrane conductance regulator; CF, cystic fibrosis.

# Introduction

Computational biomedicine integrating clinical and omics data is a modern and powerful approach for the development of novel, molecular and individualized diagnosis and treatment regimen. Specifically for organ systems, for which molecular knowledge is poor or if present was derived from animal models, computational biomedicine can give access to unexplored avenues in personalized medicine [1].

Despite rapid technological improvement and implementation to the field, the human inner ear remains a black box. When the inner ear is affected from disease, both organ functions, i.e., hearing and balance, rapidly deteriorate and in many cases do not recover. Indeed, severe hearing loss affects more than 466 million people worldwide and this number is expected to rise dramatically within the next decades [2]. State of the art treatment of hearing loss is cochlear implantation, the insertion of an electrode array into the cochlea for direct electrical activation of the auditory nerve [3]. With this approach, the damaged sensory epithelium of the cochlea, which is responsible for translating sound into electrical signals, is bypassed.

Although it is the clinically most successful neuroprosthetic device, the high inter-individual variability of therapeutic success of the cochlear implant is one of the unanswered key questions [4–6]. Indeed, a significant portion of the patients (up to 40%) experiences a less than expected hearing benefit with the device. Many factors such as age, duration of deafness, genetics, variability in cochlear anatomy, surgical technique and device characteristics, neuronal survival, electrode position or general cognitive and central processing abilities may be attributed for the variation in speech perception [7–10]. Despite the many factors that may influence the outcome of cochlear implantation, less than 20% of the variability can be actually explained and nearly none of them can be targeted therapeutically [10]. For example, patients with the same inner ear disease and implanted with the same electrode array share a wide range of outcomes [11]. Thus, outcome prediction is one of the most challenging topics in clinical cochlear implant-related research. Novel approaches in neuroscience, data analysis, molecular biology and computational analysis are required to understand the wide variability in performance amongst cochlear implant users.

For biomolecular analysis, the accessibility of the cochlea during surgical procedures offers the unique opportunity to gain a "fluid biopsy" by perilymph sampling. In previous work, we and others provided solid proof of the safety and feasibility of this method [12–15]. For example, we were able to define the cochlear microenvironment not only by analysing the proteome [16], but also the inflammasome [17] and miRNA profile [18] in human perilymph from hearing impaired patients.

Based on the idea that apart from the monogenetic disorders most of the diseases leading to hearing loss are the consequence of complex molecular changes challenging the physiological steady state of the inner ear, a single marker may not aid in the precise molecular diagnosis or even as predictor of performance. Therefore, we sought to identify comprehensive and distinctive marker profiles of the cochlear microenvironment as measured prior to implantation in patients who one year after cochlear implantation proved to be good or poor performers in speech intelligibility.

# Materials and methods

Cochlear implant recipients (n = 75 implanted ears), who were previously analysed in regard to the proteome profiles of their perilymph, were used for retrospective analysis of clinical data on hearing performance and speech intelligibility one year after cochlear implant surgery. The 45 patients who met the inclusion criteria were used to correlate the proteome profile from the perilymph analysis to the hearing performance data. Demographic data of the 45 patients are summarised in Table 1.

**Table 1. Demographic data of the patients undergoing CI surgery with perilymph sampling and one year postoperatively audiologic test data.**

| Demographic data | Age (mean in years) | n* (%) |
|---|---|---|
| Patients | 52.8 | 45 (100) |
| Male | 50.3 | 22 (48.9) |
| Female | 60 | 23 (51.1) |
| Good Performer | 53.7 | 22 (61.1) |
| Bad Performer | 50.95 | 14 (38.9) |
| Children (0–18 years) | 6 | 3 (6.7 |
| Adults (19–80) | 58.6 | 42 (91.7) |

## Perilymph sampling

Human perilymph was collected with a modified micro glass capillary during inner ear surgeries from 68 patients (75 cochleae) as already described in our previous studies [12, 19, 20]. Part of the data has been already published [12, 20]. Using modified micro glass capillary, the round window membrane was punctured directly before the insertion of a cochlear implant electrode array to obtain the perilymph samples. The protocol for collection of the perilymph samples was approved by the Ethics Committee of Hannover Medical School for perilymph by cochlear implantation (approval no. 1883–2013 and 2403–2014). Written informed consent was obtained from every patient or parent or legal guardian in case of children included in this study. The inclusion criteria for perilymph sampling is the presence of a fluid-filled cochlea as determined by magnetic resonance tomography. Every patient undergoing CI surgery was offered to participate in the study/perilymph sampling. Signed informed consent for the study and for the CI surgery was obtained. The proteome analysis was performed at Hanover Medical School.

## Proteomic analysis

In a prior study, an intraoperative perilymph sampling method and analysis by an in-depth shot-gun proteomics approach were established allowing the analysis of hundreds of proteins simultaneously in very small sample sizes in a microliters range [12]. The method for protein analysis has been published previously (e.g., in [19]). Perilymph samples were prepared for LC-MS/MS analysis by alkylation and separated using sodium dodecyl sulfate polyacrylamide gel electrophoresis as previously described [19]. Peptide samples were separated with a nanoflow ultra-high pressure liquid chromatography system (RSLC, Thermo Scientific) equipped with a trapping column (3 µm C18 particle, 2 cm length, 75 µm ID, Acclaim PepMap, Thermo Scientific) and a 50 cm long separation column (2 µm C18 particle, 75 µm ID, Acclaim Pep-Map, Thermo Scientific). The RSLC system was coupled online via a Nano Spray Flex Ion Soure II (Thermo Scientific) to an LTQ-Orbitrap Velos mass spectrometer. Metal-coated fused-silica emitters (SilicaTip, 10 µm i.d., New Objectives) and a voltage of 1.3 kV were used for the electrospray. Overview scans were acquired at a resolution of 60k in a mass range of m/z 300–1600 in the orbitrap analyser and stored in profile mode. The top 10 most intensive ions of charges two or three and a minimum intensity of 2000 counts were selected for CID fragmentation with a normalized collision energy of 38.0, an activation time of 10 ms and an activation Q of 0.250 in the LTQ. Fragment ion mass spectra were recorded in the LTQ at normal scan rate and stored as centroid m/z value and intensity pairs. Active exclusion was activated so that ions fragmented once were excluded from further fragmentation for 70 s within a mass window of 10 ppm of the specific m/z value. The relative protein quantification was performed by label-free quantification (LFQ) and was determined as LFQ intensity [19].

Additionally, proteins were subjected to classification by Gene Ontology Annotations (GOA) using *UniProt*. The Gene Ontology (GO) classification allows a mapping of the proteins into the categories *molecular function*, *biological process* and *cellular compartment*. Proteins were described using a standardized vocabulary of the *UniProt* Knowledgebase by uploading the uniprot IDs of the proteins to the *UniProt* website *http://www.uniprot.org* [19].

### Interactome analysis

All proteins obtained from MS analysis were compared for further information with the STRING database (https://string-db.org) whereas text mining was excluded using a confidence level between 0.15–0.4 enabling up to 50 interactors and full network. Candidates with the highest score were used for further comparison.

### Audiology: Classification of patients by speech intelligibility

Postoperative hearing performance with a cochlear implant one year after implantation of the 45 patients with perilymph sampling was analyzed. Therefore, data of two audiologic tests (HSM sentence test in noise at 10dB, Freiburg monosyllable word test) one year after implantation were included.

### Impedance analysis

All post-operative impedance measurements were acquired after for time points: first fitting, 3, 6 and 12 month. In order to be able to compare the impedances of different cochlear implants, the relative impedance change over time were used. For this purpose, the impedances of the individual electrodes ($E_1$ to $E_x$) of each implant were normalized to the impedance of the associated electrode at the time point of first fitting ($E_{x, \text{ time point}}$ / $E_{x, \text{ first fitting}}$).

### Statistical analysis

Mass spectrometric raw data were processed using Max Quant software (version 1.4) and human entries of Swissprot/Uniprot database. As previously described, the threshold for protein identification was set to 0.01 on peptide and protein level [19].

Proteomics data were analysed and compared by Perseus software and ingenuity pathway analysis (IPA, Qiagen Bioinformatics, http://www.ingenuity.com) software. By statistical analysis for the groups excellent and poor performers, significant differences in the levels of numerous proteins were determined. Proteins detected in at least 8 samples of a performance group and after imputation of data (replacement of missing values by normal distribution) were used for statistical analysis. Student's T-test was performed ($p < 0.05$) for identification of proteins, with significant differences in the quantification of proteins detected in the two groups.

## Results

A total of 75 perilymph samples from 68 patients were analysed. By mass spectrometry, 935 proteins were identified. The results of audiological speech intelligibility one year after cochlear implantation were available for 45 of the 68 patients. These patients were divided into two groups according to their performance data (Fig 1): Patients with good performance (n = 22) were defined with HSM sentence test in noise 10 dB > 30% and Freiburg monosyllables test > 65%, those with poor performance (n = 14) were defined with HSM sentence test in noise 10 dB < 30% and Freiburg monosyllables test < 65%. The good performer group scored a mean of 60.0% +/-18.9 on the HS + 10 dB and 78.2% +/-8.1 on the Freiburg monosyllables

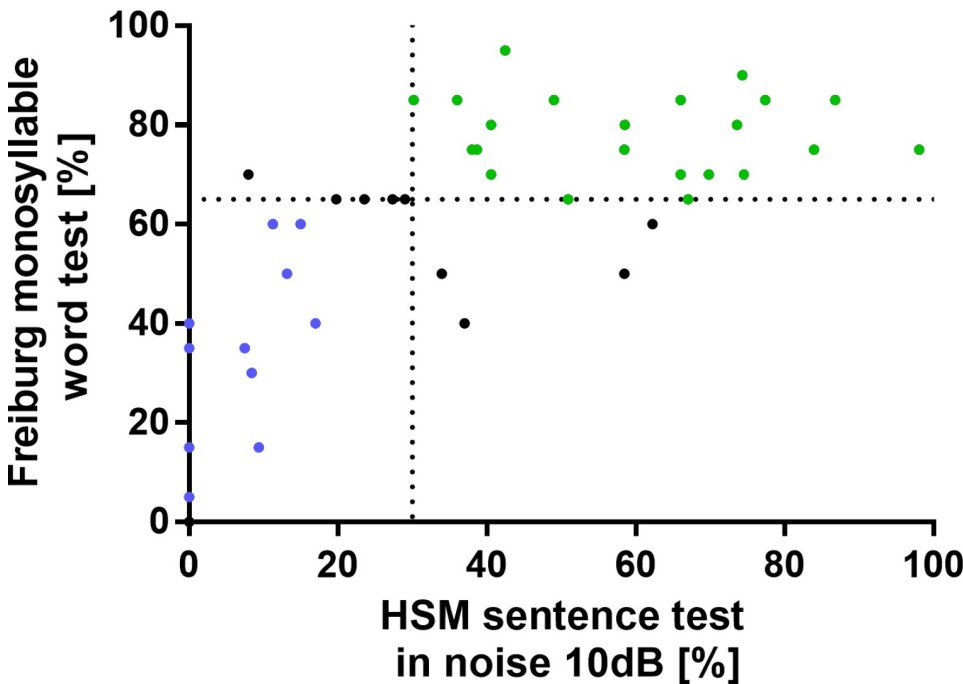

**Fig 1. Hearing performance of patients.** Patients were categorized by one year postoperative audiologic measurements into two groups. Excellent hearing performance was defined with ≥65% in Freiburger test and simultaneously ≥30% in HSM. Excellent performers (n = 22) are marked in green. Poor hearing performance was defined with <65% in Freiburger test and simultaneously <30% in HSM. Poor performers (n = 14) are depicted in blue.

test. Demographic data including aetiology are depicted in Table 2. The poor performer scored a mean of 5.8% +/-6.5 on the HSM + 10 dB and of 27.5% +/-21.7 on the Freiburg monosyllables test. Demographic data of the patients with poor performance including aetiology are depicted in Table 3. The remaining patients have not met both selection criteria and were defined as average performer. The proteome of the patients with the worse performance was then compared to the proteome of the patients with the best performance to identify differences among the groups.

**Table 2. Demographic data of good performers including aetiology.**

| Demographic data of good performers (n = 22; 61.1%) | Age (mean in years) | n* (%) |
|---|---|---|
| **Patients** | 53.7 | 22 (100) |
| **Male (n)** | 50.2 | 13 (59.1) |
| **Female (n)** | 59.5 | 9 (40.9) |
| **Children (0–18 years)** | 5.1 | 1 (4.5) |
| **Adults (19–80)** | 56.0 | 21 (95.5) |
| **Time of hearing loss before CI surgery** | 6.7 | 22 (100) |
| Etiology | | |
| **EVA** | | 1 (4,5) |
| **Menière's disease** | | 6 (27,3) |
| **Otosclerosis** | | 3 (13,6) |
| **Unknown** | | 10 (45,5) |
| **Rubella embryopathy** | | 1 (4.5) |
| **Meningitis** | | 1 (4.5) |

**Table 3. Demographic data of the patients with bad performer including aetiology.**

| Demographic data of bad performers (n = 14; 38.9) | Age (mean in years) | n* (%) |
|---|---|---|
| Patients | 51.0 | 14 (100) |
| Male (n) | 44.4 | 4 (28.6) |
| Female (n) | 53.6 | 10 (71.4) |
| Children (0–18 years) | 6.5 | 2 (14.3) |
| Adults (19–80) | 58.4 | 12 (85.7) |
| Time of hearing loss before CI surgery | 11.8 | 14 (100) |
| Etiology | | |
| EVA | | 4 (28.6) |
| Menière's disease | | 1 (7.1) |
| Otosclerosis | | 4 (28.6) |
| Unknown | | 4 (28.6) |
| CMV | | 1 (7.1) |

## Differentially expressed proteins in excellent and poor performer

The protein composition and statistically significant differences in the 2 groups were detected by relative quantification (label free quantification, LFQ intensity) of the perilymph proteins. Therefore, 287 proteins were compared, which were identified in at least eight samples in both groups (Fig 2). In the excellent hearing performance group, five proteins were identified with significantly higher abundance. In the poor hearing performance group, six proteins were identified with significantly higher abundance (Fig 2).

All proteins assigned to the cochlear signatures of excellent performer are involved in protective, inflammatory and stress regulation pathways. More specifically, extracellular regulated kinase (ERK) is activated and brain-derived neurotrophic factor (BDNF), pro-survival genes as well as antioxidative enzymes are up-regulated by the proteome signature found in patients who reached excellent performance after cochlear implantation. Each of the proteins are individually explained in the discussion. Additionally, data were analyzed by IPA. Proteins that exhibit significantly different abundances in the comparison of the two performance groups (Table 4) were further analysed by IPA software to get a functional annotation of the proteins with different abundances, visualized in diseases and functions heat maps and networks (Fig 3 and Fig 4).

Since we use different CI implants from various manufacturers, the impedances of one array to another cannot easily be compared. This is based on the individual design of the electrode array, which differs, for example, in the number of electrode contacts and thus also in their separation from basal to apical in the cochlea. In the present work, 14 different types of arrays are included and to compare the electrode impedances, the impedance values were normalized based on electrode impedances at the first fitting time point (dashed green line in Fig 5, Fig 6). Fig 5 represents the quantitative analysis of electrode impedance change at three different time points after implantation for good and bad performer. There is a wide dispersion of relative impedance changes as shown in Fig 5. When concentrating on the mean values of impedances changes, a slight increase in the impedance change can be seen from the first fitting to the follow up visits in the group of the poor performer when compared to the good performer. However, the impedances remain stable over time in both groups. The electrodes that largely coincide in their position in 14 different implants are the first basal and the last apical electrode and the relative impedance change in the most apical and the most basal electrode contact over time is depicted in Fig 6. Here again, a wide dispersion of relative impedance changes among the patients of both groups is obvious. While a slight decrease in the mean

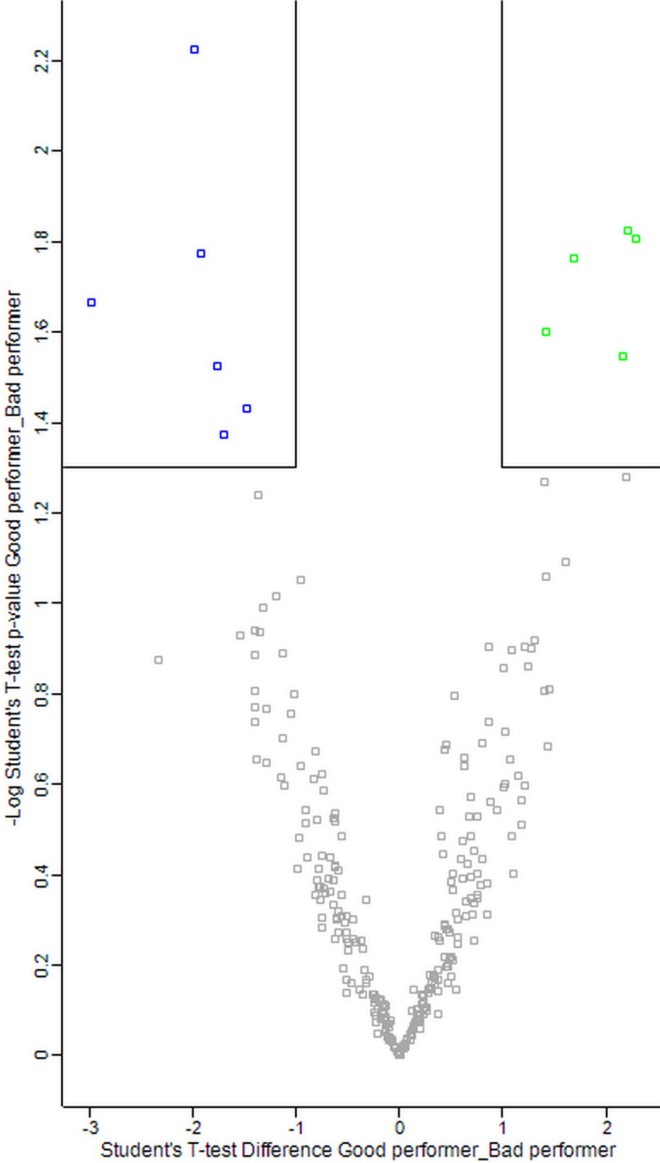

**Fig 2. Protein quantification.** Label free quantification of detected proteins in the two patients groups was performed by Max Quant software. The boxes mark the proteins with significant differences between the two hearing performance groups (p = 0.05, t-test). Marked in green are proteins significantly higher abundant in the group "excellent performers". Marked in blue are proteins significantly lower abundant in the group "poor performers".

impedance change of the apical electrode is prominent in both, the poor and good performers (Fig 6B), the impedance changes on the basal electrode increases (Fig 6A). This increase is more prominent in the group of the poor performer than in the group of the good performer.

## Discussion

This is the first report comparing human proteome data to electrophysiological audiological test results in cochlear implant recipients. With this approach, a proteome signature present in

**Table 4. Significantly higher abundant proteins in the groups excellent and poor performers.**

| Protein IDs | Protein names | Gene names | Location |
|---|---|---|---|
| P06312 | Ig kappa chain V-IV region | IGKV4-1 | Extracellular space |
| O75882 | Attractin | ATRN | Extracellular space |
| Q13228 | Selenium-binding protein 1 | SELENBP1 | Cytoplasm |
| P03952 | Plasma kallikrein | KLKB1 | Extracellular space |
| P07357 | Complement component C8 alpha chain | C8A | Extracellular space |
| P0DMV9 | Heat shock 70 kDa protein 1A/B | HSPA1B | Cytoplasm |
| P05164 | Myeloperoxidase | MPO | Cytoplasm |
| P23083 | Ig heavy chain V-I region V35 | IGHV1-2 | Other |
| P01743 | Ig heavy chain V-I region HG3 | IGHV1-46 | Other |
| A0A0C4DH24 | Immunoglobulin kappa variable 6–21 | IGKV6-21 | Other |
| P48764 | Sodium/hydrogen exchanger 3 | SLC9A3 | Plasma membrane |

Shown are significantly higher abundant proteins of the group excellent HP in green, of the group poor HP in blue. Each protein is shown in Fig 2 as single point.

patients with excellent or poor speech comprehension performance was established. Proteins, which are assigned to the to the cochlear signature of excellent performer are individually discussed below.

The selen binding protein 1 (SBP1) senses reactive xenobiotics in the cytoplasm (String database). SBP1 has a covalently binding site for selenium and is a highly conserved protein. For many cancer types, reduced expression levels of SBP1 are associated with poor survival suggesting a tumor suppressing role of SBP1 [21]. Due to its detox function, SBP1 has relevant roles in several fundamental physiological functions, from protein degradation to redox modulation. SBP1 has a significant role in the metabolism of sulfur-containing molecules and is in interaction with glutathion peroxidase GPx1 [22]. Indeed, GPx1 is highly expressed in several cell types of the inner ear including hair and supporting cells, spiral ganglion neurons, as well as cells of the stria vascularis, and is essential for maintaining cochlear homeostasis [23]. It has been proven as a molecular target for therapies addressing noise-induced hearing loss [24] and Menière's disease (clinical trial number NCT03325790). Pharmacological up-regulation of GPx1 [23] as induced by ebselen has been shown to prevent noise-induced hearing loss both in animal models as well in clinical trials [24]. On a cellular level, GPx1 up-regulation in the cochlea protects inner hair cells from denervation, prevents swelling of the afferent dendrites and of the stria vascularis and is regarded as an endogenous protective mechanism of the cochlea [23, 25, 26]. In obese patients with an unhealthy metabolic profile and at an increased risk to develop cardiovascular disease, SBP1 levels are reduced when compared to obese patients with a healthy metabolic profile [27]. Interaction database analysis reveals that SBP1 is in contact with PLAUR, which encodes the urokinase receptor (uPAR) [27]. This, in turn, promotes cell survival.

Attractin is encoded by the ATRN gene, which has many transcript variants. One isoform is a secreted protein involved in the initial immune cell clustering during inflammatory responses to regulate the chemotactic activity of chemokines [28]. It shows interaction with MGRN1, an E3 ubiquitin ligase. Membrane-bound attractin is anchored on the surface of neurons or glial cells and mediates the myelination signal through its extracellular domains [29]. Secreted attractin may interfere with membrane-bound attractin thereby disrupting neurite formation in differentiating cortical neural cells in vitro [30]. Interestingly, secreted attractin that is present in the circulation is prevented by the blood-brain-barrier to enter the CNS and to interfere with the membrane-bound form [30]. For the brain, attractin has been shown to

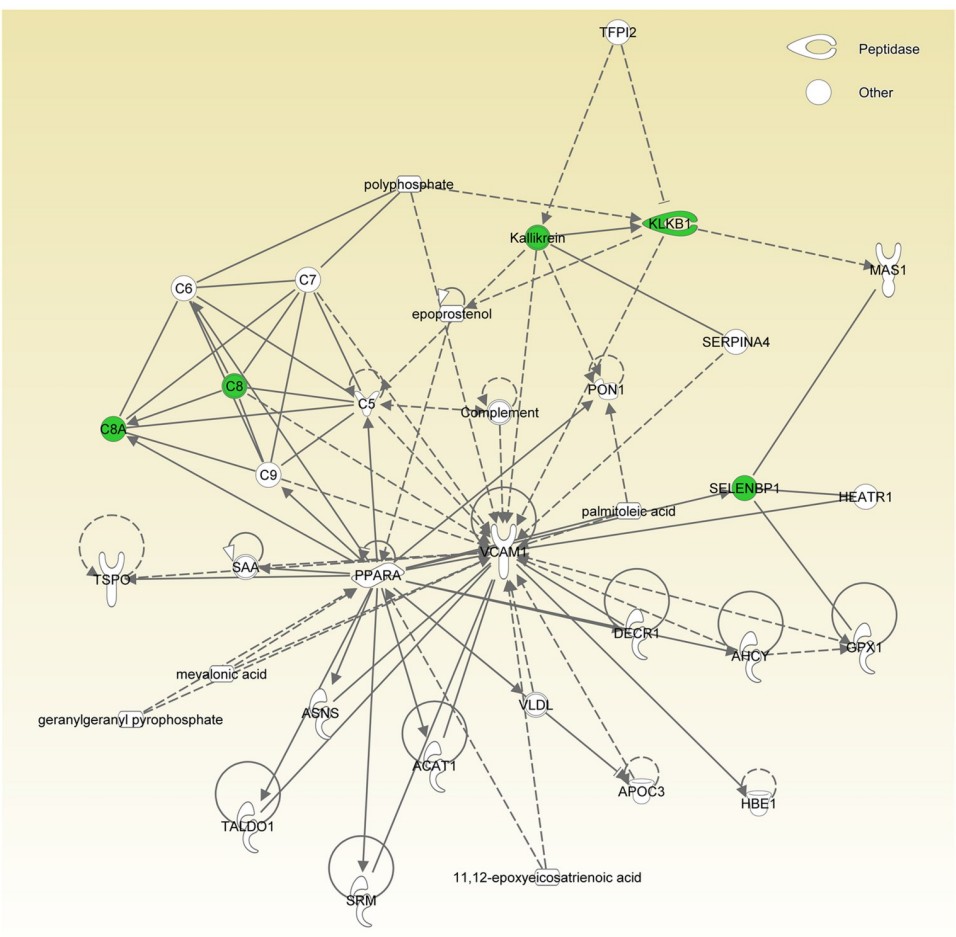

**Fig 3. Networks of higher abundant proteins in the group excellent hearing performance.** Higher abundant proteins identified in the two hearing performance groups were uploaded and analyzed by IPA software. Shown are the main networks in which the proteins are involved in. Shown is the network in which all 5 higher abundant proteins (marked in green) in the group excellent hearing performance are involved. The function of this network is described by IPA software with Developmental Disorder, Hereditary Disorder, Immunological Disease.

play a neuroprotective role [31]. The role of attractin in the perilymph is unclear. However, since lack of attractin is associated with oxidative stress and neurodegeneration by a decrease of extracellular regulated kinase (ERK), attractin might be involved in mediating cell survival under oxidative stress [32]. The role of ERK-mediated neuroprotection in auditory neurons in vitro [33] and in vivo [34] is leading to the hypothesis that increased attractin levels in the perilymph of excellent performer might be related to an improved cochlear health at the time of implantation. However, there is no evidence whatsoever to support this hypothesis and the role of attractin in the inner ear needs further investigation.

Another factor found in the perilymph of patients that later were shown to become excellent performer with their implant is complement C8, an important antibacterial immune effector [35]. Complement C8 initiates membrane penetration and coordinates membrane attack complex (MAC) pore formation leading to cell lysis [35]. It has close interaction with other complement types e.g., C6, C7, C9. However, at concentrations that are considered sublytic, bound C5b–C9 complex activates protective intracellular signaling pathways induce resistance to apoptosis and upregulation of pro-survival genes such as Bcl-2 [36].

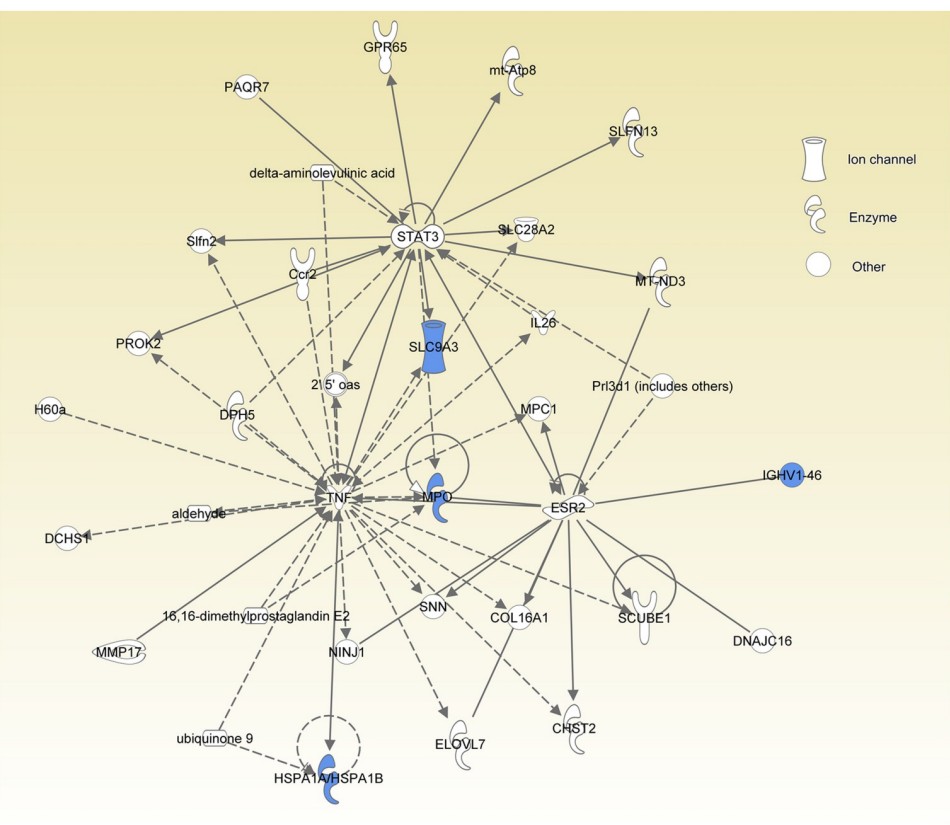

**Fig 4. Networks of higher abundant proteins in the group of the poor hearing performer.** Higher abundant proteins identified in the two hearing performance groups were uploaded and analyzed by IPA software. Shown are the main networks in which the proteins are involved in. Shown is the network in which 5 of 6 higher abundant proteins (marked in blue) in the group poor hearing performance are involved. The function of this network is described by IPA software with Carbohydrate Metabolism, Molecular Transport, Small Molecule Biochemistry.

Kallikreins are serine proteases that cleave kininogens to produce bradykinin leading to inflammation. Kallikrein 1 triggers sensory nerve stimulation [37]. In addition, kallikreins enable kinin signaling thereby promoting cell survival, reducing oxidative stress and maintaining cellular integrity [38]. Tissue kallikreins increase the expression of the neurotrophin brain-derived neurotrophic factor (BDNF) and of pro-survival Bcl-2 genes [38]. Due to its neuroprotective effect, the biomacromolecule kallikrein is specifically of interest in treating cerebral ischemia injury [39].

The immunoglobulin chain IGKV4–1 was found to be up-regulated in the patients with excellent performance, whereas IGHV1-2, IGHV1-46 and IGKV6-21 were up-regulated in patients with poor performance. Interestingly, when comparing demyelinating with remyelinating lesions of the central nervous system, the IGHV4-1 transcript that is also found in the B cell receptor transcriptome of the CSF was up-regulated [40].

Myeloperoxidase (MPO) is synthesized in promyelocytes and promyelomonocytes in the bone marrow [41]. Belonging to the haem peroxidase-cyclooxygenase superfamily, MPO can oxidize tyrosine to tyrosyl radical or chloride to hypochlorous acid (HOCl) using hydrogen peroxide as an oxidizing agent. Tyrosyl radical, a cytotoxic molecule, is released by neutrophils to kill bacteria and other pathogens [41]. Thus, patients with MPO deficiencies have a significant risk for infections [42]. On the other hand, increased levels of oxidants by MPO can

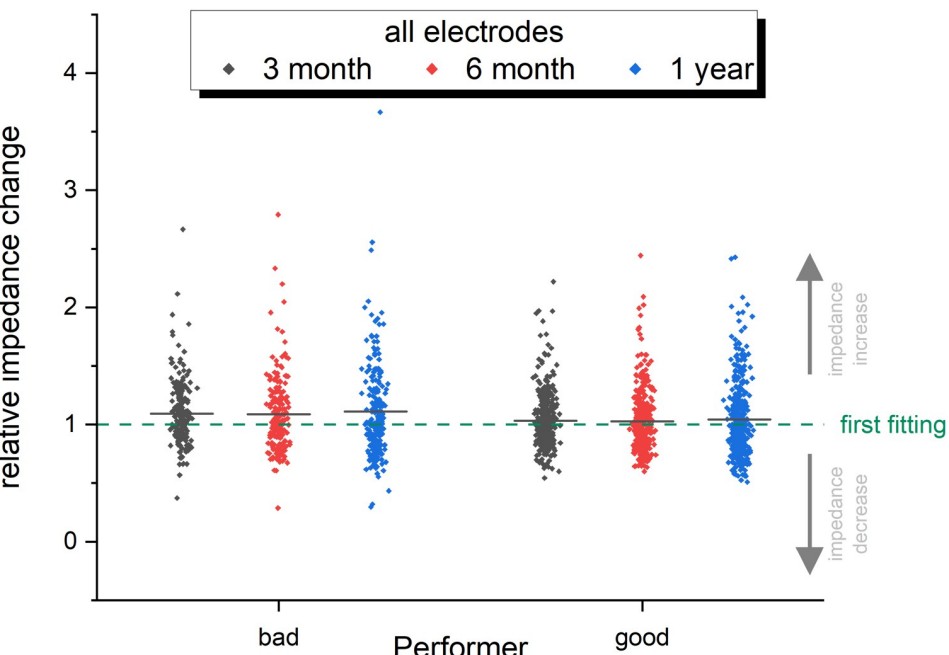

**Fig 5. Relative impedance change (all electrodes).** Relative impedance change of 14 different electrode arrays of bad and good performers normalized on electrode impedances at first fitting time point (dashed line) for 3 time points after implantation (3 month: black; 6 month: red; 1 year: blue dots).

damage tissue damage and are found in many diseases characterized by acute or chronic inflammation [41] such as cardiovascular and neurodegenerative diseases. Inflammation-associated oxidants such as HOCl-and tyrosyl radicals may bind to the components of the extracellular matrix and to proteins leading to the changes associated with oxidant damage such as atherosclerosis [43]. In animal models, neutrophil-derived MPO seems to be involved in the pathogenesis of Alzheimer's like disease and inhibition of MPO might present a novel therapeutic target to combat cognitive decline [44]. Indeed, MPO-derived HOCl might induce senescence [45]. Also, MPO activity is increased in a model of graft versus host disease and is increased in the senescence-associated secretory phenotype [46]. In the inner ear, expression of MPO is associated with cochlear dysfunction probably due to MPO-catalyzed ROS accumulation and damage of the stria vascularis [47]. Whether and how MPO and poor performance after cochlear implantation might be linked is not known and needs further investigation.

Heat shock proteins such as Hsp70 have diverse functions in protein folding and restoration. For the inner ear, a protective effect of Hsp70 has been shown in several experimental models [48–50]. However, Hsp70 is also a stress marker when unfolded proteins accumulate due to proteotoxic stress [51]. In addition, Hsp70 interacts with BAG5, a chaperone regulator, to reduce ubiquitination of the client protein STIP1, a stress-induced phosphoprotein and co-chaperone of Hsp90 [52]. Cells lacking STIP1 compensate the proteasomal defect by improved protein folding [53], thereby reequilibrating the proteostatic balance in diseases such as neurodegenerative disorders [53].

The solute carrier family 9, subfamily A, member 3 (SLC9A3, NHE-3) is a sodium–hydrogen exchanger (NHE) that acts in pH regulation and plays a role in signal transduction to form a chemical gradient of ions for absorbing the sodium ion. In the gerbil inner ear, 4 isoforms of NHE, including NHE-3 are expressed. Although the other isoforms are broadly expressed in the inner ear, NHE-3 expression was limited to the apical surface of the marginal

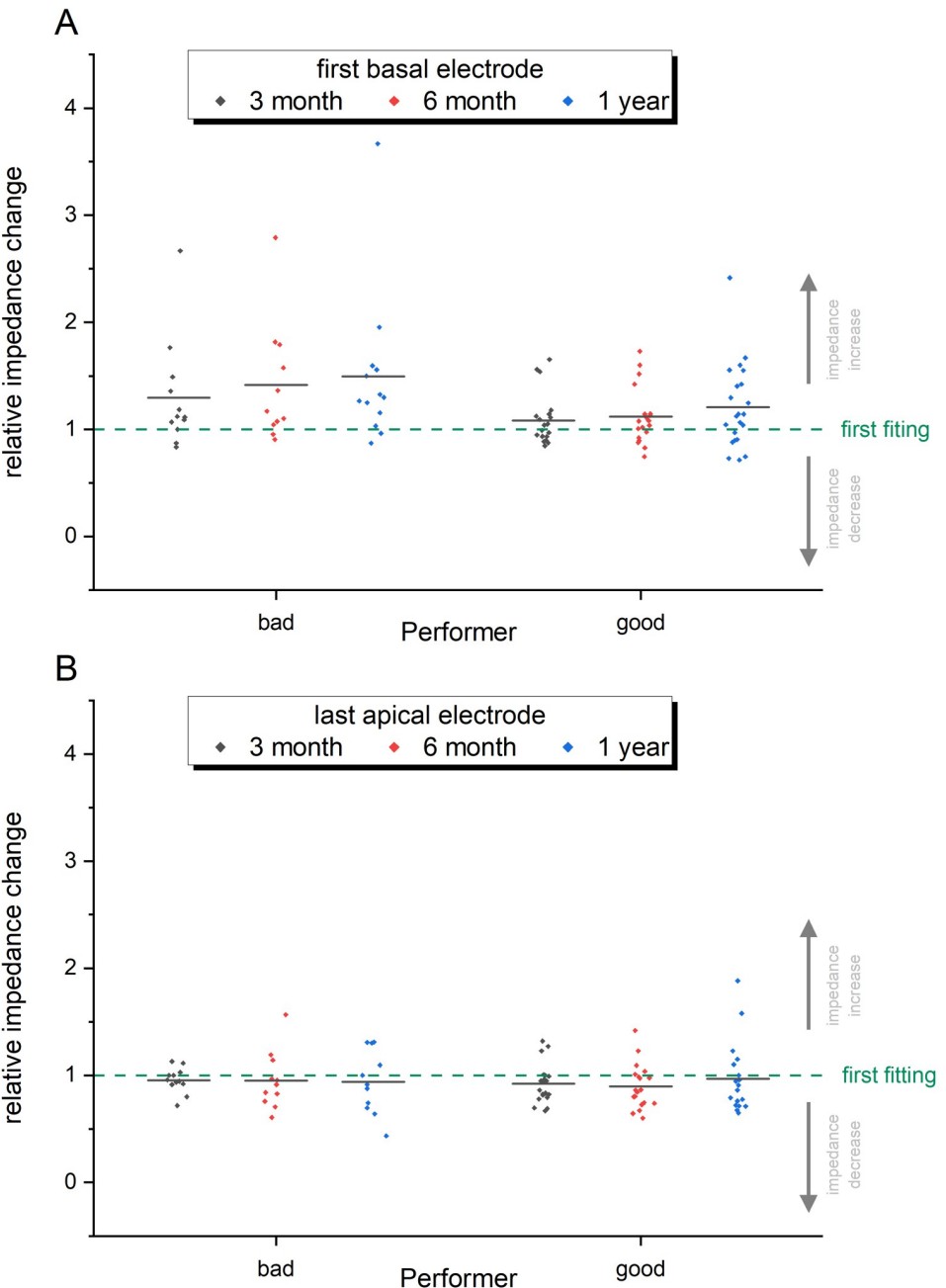

**Fig 6. Relative impedance change (first basal / last apical electrode).** Relative impedance change of 14 different electrode arrays of bad and good performers normalized on electrode impedances at first fitting time point (dashed line) for 3 time points after implantation (3 month: black; 6 month: red; 1 year: blue dots). A) First basal electrode; B) Last apical electrode.

cells in the stria vascularis [54]. Based on data from the interaction database STRING, NHE-3 is linked with Rho kinase, endothelin-1, calcineurin, calmodulin signaling pathways, all pathways playing a crucial role in inner ear homeostasis and pathophysiology and presenting interesting drug targets [55–57]. For example, calcineurin activation contributes to noise induced hearing loss [57]. Endothelin induces vasospasms of the spiral modiolar artery via activation of

the Rho-kinase and an increase in calcium-sensitivity [55]. Another role of SLC9A3 as an important interacting partner or modifier of the cystic fibrosis transmembrane conductance regulator (CFTR), a gene of which mutations are causative for cystic fibrosis (CF), has been described recently [58]. SLC9A3 gene variants can affect susceptibility to bacterial infections and severity of pulmonary condition by interaction with CFTR, thus providing evidence of its modular effect on CF [58]. Whether and how gene variants of SLC9A3 could affect genes important for cochlear homeostasis is unknown hitherto.

The present work is a pilot study and shows how proteome data can be linked to functional data in cochlear implantation and correlated using artificial intelligence-based bioinformatics analysis approaches. Whether the determined molecular perilymph profile can be used as a predictive factor for individual implant performance cannot be concluded from the present results. The main limitation of the study is the time gap between determining the proteome profile of the cochlea and the outcome data of cochlear implantation. Lack of methods to collect perilymph at time intervals such as weeks or months after implantation make it impossible to correlate speech performance data to the actual perilymph profile. Thus, inflammation processes especially related to the surgery and implantation procedure were not taken into account. In the present study, we also analysed the presence of residual hearing and impedance values post implantation as additional factors that influence the outcome after implantation. However, due to the heterogeneous patient population and the different implants used, a validated association cannot be derived from the results. We merely compared the proteome profile at the time point of implantation and correlated to the later outcome as measured by speech perception tests. Whether perilymph profiles can be used as predictive factors for speech performance after cochlear implantation needs verification in multicentre and controlled studies including a higher number of patients. This is especially of importance in order to control for the different clinical factors that have been shown to predict speech perception outcome in adult cochlear implant recipients [59]. Even so, our data allow the cautious assumption that in the near future, therapeutic decision trees such as drug device combinations for patients suffering from acute or chronic inner ear disease may emerge depending on their perilymph molecular profile.

## Conclusion

Statistical analysis of the perilymph proteome identified significantly elevated differential proteins in both patient groups, i.e., patients with excellent and patients with poor performance after cochlear implantation. This could open up a new possibility to make statements on the pathophysiology of the hearing disorder and to predict the individual hearing performance of patients after cochlear implantation by means of a molecular perilymph analysis or to be able to add a special local drug therapy in order to improve the performance.

## Author Contributions

**Conceptualization:** Athanasia Warnecke, Thomas Lenarz, Heike Schmitt.

**Data curation:** Martin Durisin, Caroline Krüger, Melanie Steffens, Heike Schmitt.

**Formal analysis:** Caroline Krüger, Andreas Pich, Athanasia Warnecke, Melanie Steffens, Heike Schmitt.

**Funding acquisition:** Thomas Lenarz.

**Investigation:** Andreas Pich, Athanasia Warnecke.

**Methodology:** Carsten Zeilinger.

**Project administration:** Athanasia Warnecke.

**Resources:** Martin Durisin, Athanasia Warnecke, Nils Prenzler.

**Validation:** Carsten Zeilinger.

**Writing – original draft:** Caroline Krüger, Athanasia Warnecke.

**Writing – review & editing:** Martin Durisin, Melanie Steffens, Carsten Zeilinger, Thomas Lenarz, Nils Prenzler, Heike Schmitt.

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
