## [Decision Letter · Decision Letter 0]

24 May 2021

PONE-D-21-08371

Proteome profile of patients with excellent and poor speech intelligibility after cochlear implantation: Can perilymph proteins predict performance?

PLOS ONE

Dear Dr. Warnecke,

Thank you for submitting your manuscript to PLOS ONE. After careful consideration, we feel that it has merit but does not fully meet PLOS ONE’s publication criteria as it currently stands. Therefore, we invite you to submit a revised version of the manuscript that addresses the points raised during the review process.

The reviewers have both agreed that the manuscript provides interesting and important data on the proteome of the inner ear ear in cochlear implant patients.  They provided comments on the manuscript and have raised several points, particularly regarding the discussion and interpretation of the findings that you should consider.  

We look forward to receiving your revised manuscript.

Kind regards,

Peter Rowland Thorne, CNZM PhD

Academic Editor

PLOS ONE

Journal Requirements:

3. In your Methods section, please provide additional information about the participant recruitment method and the demographic details of your participants. Please ensure you have provided sufficient details to replicate the analyses such as: a) a description of any inclusion/exclusion criteria that were applied to participant recruitment, b) a description of how participants were recruited, and c) descriptions of where participants were recruited and where the research took place.

4. Please ensure your Methods and reagents are described in sufficient detail for another researcher to reproduce the experiments described. Specifically, please ensure you have provided a brief description of perlymph sample preparation, or an appropriate reference for standard methods.

6. We noticed you have some minor occurrence of overlapping text with the following previous publication(s), which needs to be addressed:

- "Detection of BDNF-Related Proteins in Human Perilymph in Patients with Hearing Loss," doi: 10.3389/fnins.2019.00214

In your revision ensure you cite all your sources (including your own works), and quote or rephrase any duplicated text outside the methods section. Further consideration is dependent on these concerns being addressed.

Reviewers' comments:

Reviewer's Responses to Questions

**Comments to the Author**

1. Is the manuscript technically sound, and do the data support the conclusions?

Reviewer #1: Partly

Reviewer #2: Yes

2. Has the statistical analysis been performed appropriately and rigorously? 

Reviewer #1: I Don't Know

Reviewer #2: Yes

3. Have the authors made all data underlying the findings in their manuscript fully available?

Reviewer #1: Yes

Reviewer #2: Yes

4. Is the manuscript presented in an intelligible fashion and written in standard English?

Reviewer #1: Yes

Reviewer #2: Yes

5. Review Comments to the Author

Reviewer #1: Cochlear implantation provides a unique opportunity to sample human perilymph in a safe and ethically appropriate way. The authors of this paper have previously used this sampling technique to characterise the protein composition or proteotome in order to give an insight into the composition of proteins able to be sampled from perilymph. This is potentially a very important technique for ongoing research into inner ear pathology. This study was performed retrospectively to compare the protein profiles in different Cochlear implant candidates with their performance a year after the surgery. It would certainly appears that this study was conducted as an afterthought.

As is mentioned, there are significant differences in hearing performance among Cochlear implant candidates, approximately 20% of which cannot be explained. Research into this issue is certainly welcomed. There a number of issues with this study which mean that it does little to advance our ability to answer these questions, other than suggesting the sort of investigation as a possible avenue of research. There are relatively few patients that can be matched via implant performance and proteotome, the final protein analysis involves 287 proteins and at least 8 samples in excellent and poor performing groups. Both adults and children are used which would likely increase the heterogenicity of causes of hearing loss. By necessity, the perilymph samples are only taken before electrode insertion, thus do not reflect protein changes due to the insertional trauma and subsequent inflammation. There is certainly no getting around this and this is acknowledged by the authors.

Different individual reactions to the combination of electrode insertion or trauma and foreign body reaction to the electrode are likely to be part of the variations in performance that we see. It would have been useful to know whether the was residual hearing lost in these people. It would also be useful to have data regarding electrode impedance, which may act as an indication of the extent of fibrosis around the electrode. It is not stated in the paper, but probably the patients used in the study did not have substantial residual hearing.

Overall, while I do not think any conclusions can be made about the results of this study, Due to the very early stages of this type of research, it is of interest to other groups and may stimulate further research along these same lines.

Reviewer #2: This is a nice and innovative paper.

Please increase the resolution of figures 3 and 4, as they now appear blurry.

More in-depth discussion is needed for the proteins that were changed in each group, including some speculation on why and how these are related to the outcomes (poor or good).

It would be helpful to add to the discussion some details on how the correlation might help the process of patient selection or additional treatments in the clinic.

6. PLOS authors have the option to publish the peer review history of their article (what does this mean?). If published, this will include your full peer review and any attached files.

Reviewer #1: **Yes: **Philip Antony Bird

Reviewer #2: **Yes: **Yehoash Raphael

---

## [Author Response · Author response to Decision Letter 0]

16 Dec 2021

Reviewers' comments:

Reviewer's Responses to Questions

Comments to the Author

1. Is the manuscript technically sound, and do the data support the conclusions?

Reviewer #1: Partly 

ANTWORT

Reviewer #2: Yes

2. Has the statistical analysis been performed appropriately and rigorously? 

Reviewer #1: I Don't Know

ANTWORT

Reviewer #2: Yes

3. Have the authors made all data underlying the findings in their manuscript fully available?

Reviewer #1: Yes

Reviewer #2: Yes

4. Is the manuscript presented in an intelligible fashion and written in standard English?

Reviewer #1: Yes

Reviewer #2: Yes

5. Review Comments to the Author

Reviewer #1:

a) Cochlear implantation provides a unique opportunity to sample human perilymph in a safe and ethically appropriate way. The authors of this paper have previously used this sampling technique to characterise the protein composition or proteotome in order to give an insight into the composition of proteins able to be sampled from perilymph. This is potentially a very important technique for ongoing research into inner ear pathology. This study was performed retrospectively to compare the protein profiles in different Cochlear implant candidates with their performance a year after the surgery. It would certainly appears that this study was conducted as an afterthought.

Response: Thank you for your kind comment. The reviewer is right; the proteome data have been previously published. However, a correlation of the proteome profile at the time point of implantation and the speech performance data with the implant has not been performed and was the focus of the present study.

b) As is mentioned, there are significant differences in hearing performance among Cochlear implant candidates, approximately 20% of which cannot be explained. Research into this issue is certainly welcomed. There a number of issues with this study which mean that it does little to advance our ability to answer these questions, other than suggesting the sort of investigation as a possible avenue of research.

Response: As stated by the reviewer, there are some issues related with our study. The number of the patients included is limited, the changes of the proteome associated with the implantation procedure cannot be, at least with the currently available technology, determined. Thus, the study shows that there might be a possibility that the state in which the CI is implanted might determine outcomes and that perilymph analysis might provide an interesting predictive tool. The limitations as also acknowledged by the reviewer are discussed in the manuscript. For more clarity, we have revised this section as follows.

Line 387-390: “The present work is a pilot study and shows how proteome data can be linked to functional data in cochlear implantation and correlated using artificial intelligence-based bioinformatics analysis approaches. Whether the determined molecular perilymph profile can be used as a predictive factor for individual implant performance cannot be concluded from the present results.” 

c) There are relatively few patients that can be matched via implant performance and proteotome, the final protein analysis involves 287 proteins and at least 8 samples in excellent and poor performing groups. Both adults and children are used which would likely increase the heterogenicity of causes of hearing loss. By necessity, the perilymph samples are only taken before electrode insertion, thus do not reflect protein changes due to the insertional trauma and subsequent inflammation. There is certainly no getting around this and this is acknowledged by the authors.

Response: Thank you for your kind comment. This shows how difficult research in this field can be due to limitations in the accessibility of the inner ear. 

d) Different individual reactions to the combination of electrode insertion or trauma and foreign body reaction to the electrode are likely to be part of the variations in performance that we see. It would have been useful to know whether the was residual hearing lost in these people.

Response: In our previous study (Schmitt et al 2017), we have analysed residual hearing of patients to show whether perilymph sampling leads to loss of residual hearing. Additionally, we included audiology and proteomics data of patients of our recent study (Personalized Proteomics for Precision Diagnostics in Hearing Loss: Disease-Specific Analysis of Human Perilymph by Mass Spectrometry; Heike A. Schmitt*, Andreas Pich, Nils K. Prenzler, Thomas Lenarz, Jennifer Harre, Hinrich Staecker, Martin Durisin, and Athanasia Warnecke; ACS Omega 2021, 6, 33, 21241–21254; Publication Date:August 13, 2021; https://doi.org/10.1021/acsomega.1c01136). 

After interpretation of the pure tone audiograms before cochlear implantation of the study patients relating to any residual hearing, we could evaluate following results: Of the 14 patients with bad performance, 11 patients had profound hearing loss and 3 patients showed some residual hearing. Of the 22 patients with good performance 12 patients had profound hearing loss and 10 patients showed some residual hearing. The results show that most of the patients with bad performance had preoperatively no residual hearing; however, nearly half of the good performance patients group had also preoperatively no residual hearing. 

Although the presence of residual hearing is acknowledged as one of the known factors to influence performance with a CI, there must be more factors affecting the outcome, since not all patients without residual hearing are bad performers and not all patients with residual hearing are good performer. Whether the proteome profile alone or in combination with other factors such as residual hearing may be of use to predict poor and good performer needs further analysis. With the limited numbers from our study we simply can show that there are distinct profiles among some patient groups and that this finding needs further investigation.

e) It would also be useful to have data regarding electrode impedance, which may act as an indication of the extent of fibrosis around the electrode. It is not stated in the paper, but probably the patients used in the study did not have substantial residual hearing.

Response: Thank you for this interesting suggestion. We initially refrained from correlating perilymph data to impedances due to the heterogeneity of the implants used. As suggested by the reviewer, we (our new co-author Dr. Melanie Steffens) have performed this analysis. She also looked whether the good or bad performance was related to a specific electrode type and position in the cochlea as cofounding factor. Since the group included in the present analysis was heterogenous in terms of the implant manufacturer as well as the electrode and implant type, this analysis did not reveal any additional results to the present study. We have added the results to the manuscript and to the discussion one sentence acknowledging this fact. 

Line 248-266: “Since we use different CI implants from various manufacturers, the impedances of one array to another cannot easily be compared. This is based on the individual design of the electrode array, which differs, for example, in the number of electrode contacts and thus also in their separation from basal to apical in the cochlea. In the present work, 14 different types of arrays are included and to compare the electrode impedances, the impedance values were normalized based on electrode impedances at the first fitting time point (dashed green line in Fig 5, Fig 6). Figure 5 represents the quantitative analysis of electrode impedance change at three different time points after implantation for good and bad performer. There is a wide dispersion of relative impedance changes as shown in Fig. 5. When concentrating on the mean values of impedances changes, a slight increase in the impedance change can be seen from the first fitting to the follow up visits in the group of the poor performer when compared to the good performer. However, the impedances remain stable over time in both groups. The electrodes that largely coincide in their position in 14 different implants are the first basal and the last apical electrode and the relative impedance change in the most apical and the most basal electrode contact over time is depicted in Figure 6. Here again, a wide dispersion of relative impedance changes among the patients of both groups is obvious. While a slight decrease in the mean impedance change of the apical electrode is prominent in both, the poor and good performer (Fig 6 B), the impedance changes on the basal electrode increases (Fig 6 A). This increase is more prominent in the group of the poor performer than in the group of the good performer.

Line 395-399: “In the present study, we also analysed the presence of residual hearing and impedance values post implantation as additional factors that influence the outcome after implantation. However, due to the heterogenous patient population and the different implants used, a validated association cannot be derived from the results.” 

Line 268-276:

 “Figure 5: Relative impedance change (all electrodes)

Relative impedance change of 14 different electrode arrays of bad and good performers normalized on electrode impedances at first fitting time point (dashed line) for 3 time points after implantation (3 month: black; 6 month: red; 1 year: blue dots).

Figure 6: Relative impedance change (first basal / last apical electrode)

Relative impedance change of 14 different electrode arrays of bad and good performers normalized on electrode impedances at first fitting time point (dashed line) for 3 time points after implantation (3 month: black; 6 month: red; 1 year: blue dots). A) First basal electrode; B) Last apical electrode.”

Overall, while I do not think any conclusions can be made about the results of this study, Due to the very early stages of this type of research, it is of interest to other groups and may stimulate further research along these same lines.

Response: Thank you for your comment. As stated in the manuscript, we do not draw any conclusions form the molecular profiles. We simply observed these two different profiles in the two different groups of patients and discussed each protein individually. Whether and how the proteome profile influences implant performance needs further research. As stated by the reviewer, we simply wanted to delineate this avenue and stimulate research along this line. We hope that this now becomes much clearer from reading the revised version of the manuscript. 

Reviewer #2: This is a nice and innovative paper.

a) Please increase the resolution of figures 3 and 4, as they now appear blurry.

Response: Thank you for your kind comment. The figures have been revised and replaced accordingly. 

b) More in-depth discussion is needed for the proteins that were changed in each group, including some speculation on why and how these are related to the outcomes (poor or good).

Response: We have initially thought to discuss the proteins in more detail and also to include speculations why and how these could relate to cochlear health and implant performance. However, we refrained from doing so. Especially in lieu with the comments of reviewer 1 we would think that such an in- depth discussion would be misleading. We hope to be able to initiate worldwide perilymph analysis and thus be able to generate more homogenous groups for improved analysis. Indeed, we have already started international collaborations to this aim. However, where suitable, we have emphasized the neuroprotective potential of the different molecules since we speculate that specific factors may be released upon damage / hearing loss in the inner ear that serve to preserve neuronal function alongside the auditory pathway. This is very speculative and thus we have not included this speculation to the manuscript. 

c) It would be helpful to add to the discussion some details on how the correlation might help the process of patient selection or additional treatments in the clinic.

Response: Thank you for your interesting suggestion. It is indeed an excellent and futuristic suggestion to use molecular correlations to performance for patient selection and additional treatments. Unfortunately, as stated in the answers above, such a speculation, as interesting and important it may be, would be too early and cannot be made based on our preliminary results. As stated above: However, where suitable, we have emphasized the neuroprotective potential of the different molecules since we speculate that specific factors may be released upon damage / hearing loss in the inner ear that serve to preserve neuronal function alongside the auditory pathway. This is very speculative and thus we have not included this speculation to the manuscript. 

6. PLOS authors have the option to publish the peer review history of their article (what does this mean?). If published, this will include your full peer review and any attached files.

Do you want your identity to be public for this peer review? For information about this choice, including consent withdrawal, please see our Privacy Policy.

Reviewer #1: Yes: Philip Antony Bird

Reviewer #2: Yes: Yehoash Raphael 

Journal Requirements:

Response: We checked the references for correctness. We added 5 new references and refreshed one:

Reference list number 2 (refreshed)

Reference list number 19

Reference list number 20

Reference list number 31

Reference list number 39

Reference list number 40

Response: We made some formatting changes due to the style requirements of the journal. We changed the manuscript into grouped style and integrated the figures in the running text.

Response: n the case of children undergoing CI surgery and perilymph sampling we obtained the consent of parent or legal guardian. We added this information in the methods section.

3. In your Methods section, please provide additional information about the participant recruitment method and the demographic details of your participants. Please ensure you have provided sufficient details to replicate the analyses such as: a) a description of any inclusion/exclusion criteria that were applied to participant recruitment, b) a description of how participants were recruited, and c) descriptions of where participants were recruited and where the research took place.

Response: The demografic data of the study participants are depicted in detail in Table 1 – 3. The Tables were now integrated in the running text as required in the style requirements of the journal.

Additionally we included following information in the methods section:

The inclusion criteria for perilymph sampling is the planned procedure of a CI surgery. Every patient undergoing CI surgery has the ability to participate in the study/perilymph sampling and to sign the consent form for the study in parallel to the consent form for the CI surgery. The only exclusion criteria is the existence of any infectious disease of the patient or the impossibility of perilymph sampling due to the individual surgical procedure decided by the surgeon. The proteome analysis takes place in the Hanover Medical School.

4. Please ensure your Methods and reagents are described in sufficient detail for another researcher to reproduce the experiments described. Specifically, please ensure you have provided a brief description of perlymph sample preparation, or an appropriate reference for standard methods.

Response: In the methods section we refer to our previous publication (Proteome Analysis of Human Perilymph using an Intraoperative Sampling Method. Journal of Proteome Research. 2017 Mar 10;acs.jproteome.6b00986; Schmitt et al.). In this publication an detailed description of the sampling method, perilymph sample preparation and analysis is included.

6. We noticed you have some minor occurrence of overlapping text with the following previous publication(s), which needs to be addressed:

Response: - "Detection of BDNF-Related Proteins in Human Perilymph in Patients with Hearing Loss," doi: 10.3389/fnins.2019.00214

In your revision ensure you cite all your sources (including your own works), and quote or rephrase any duplicated text outside the methods section. Further consideration is dependent on these concerns being addressed.

Response: We added the missing citations in the methods section “proteomic analyses” and “statistical analyses”

---

## [Decision Letter · Decision Letter 1]

27 Jan 2022

Proteome profile of patients with excellent and poor speech intelligibility after cochlear implantation: Can perilymph proteins predict performance?

PONE-D-21-08371R1

Dear Dr. Warnecke,

We’re pleased to inform you that your manuscript has been judged scientifically suitable for publication and will be formally accepted for publication once it meets all outstanding technical requirements. 

Kind regards,

Peter Rowland Thorne, CNZM PhD

Academic Editor

PLOS ONE

Additional Editor Comments (optional):

All comments from reviewers and editors have been addressed

Reviewers' comments:

Reviewer's Responses to Questions

**Comments to the Author**

1. If the authors have adequately addressed your comments raised in a previous round of review and you feel that this manuscript is now acceptable for publication, you may indicate that here to bypass the “Comments to the Author” section, enter your conflict of interest statement in the “Confidential to Editor” section, and submit your "Accept" recommendation.

Reviewer #1: All comments have been addressed

2. Is the manuscript technically sound, and do the data support the conclusions?

Reviewer #1: Yes

3. Has the statistical analysis been performed appropriately and rigorously? 

Reviewer #1: Yes

4. Have the authors made all data underlying the findings in their manuscript fully available?

Reviewer #1: Yes

5. Is the manuscript presented in an intelligible fashion and written in standard English?

Reviewer #1: Yes

6. Review Comments to the Author

Reviewer #1: I'm satisfied that all of my suggestions and comments have been addressed. I am therefore happy for the paper to be accepted for publication.

7. PLOS authors have the option to publish the peer review history of their article (what does this mean?). If published, this will include your full peer review and any attached files.

Reviewer #1: **Yes: **Philip Antony Bird

---

## [Editor Report · Acceptance letter]

24 Feb 2022

PONE-D-21-08371R1 

Proteome profile of patients with excellent and poor speech intelligibility after cochlear implantation: Can perilymph proteins predict performance? 

Dear Dr. Warnecke:

I'm pleased to inform you that your manuscript has been deemed suitable for publication in PLOS ONE. Congratulations! Your manuscript is now with our production department. 

Kind regards, 

on behalf of

Dr. Peter Rowland Thorne 

Academic Editor

PLOS ONE